# Anti-Adipogenic Effects of Delphinidin-3-*O*-*β*-Glucoside in 3T3-L1 Preadipocytes and Primary White Adipocytes

**DOI:** 10.3390/molecules24101848

**Published:** 2019-05-14

**Authors:** Miey Park, Anshul Sharma, Hae-Jeung Lee

**Affiliations:** Department of Food and Nutrition, Gachon University, Gyeonggi-do 13120, Korea; mieyp@naver.com (M.P.); anshul.silb18@gmail.com (A.S.)

**Keywords:** anthocyanin, delphinidin-3-*O*-*β*-glucoside, 3T3-L1 adipocytes, primary white adipocytes, adipogenesis, differentiation

## Abstract

Delphinidin-3-*O*-*β*-glucoside (D3G) is a health-promoting anthocyanin whose anti-obesity activity has not yet been thoroughly investigated. We examined the effects of D3G on adipogenesis and lipogenesis in 3T3-L1 adipocytes and primary white adipocytes using real-time RT-PCR and immunoblot analysis. D3G significantly inhibited the accumulation of lipids in a dose-dependent manner without displaying cytotoxicity. In the 3T3-L1 adipocytes, D3G downregulated the expression of key adipogenic and lipogenic markers, which are known as peroxisome proliferator-activated receptor gamma (PPARγ), sterol regulatory element-binding transcription factor 1 (SREBP1), CCAAT/enhancer-binding protein alpha (C/EBPα), and fatty acid synthase (FAS). Moreover, the relative protein expression of silent mating type information regulation 2 homolog 1 (SIRT1) and carnitine palmitoyltransferase-1 (CPT-1) were increased, alongside reduced lipid levels and the presence of several small lipid droplets. Furthermore, D3G increased the phosphorylation of adenosine monophosphate-activated protein kinase (AMPK) and acetyl-CoA carboxylase (ACC), which suggests that D3G may play a role in AMPK and ACC activation in adipocytes. Our data indicate that D3G attenuates adipogenesis and promotes lipid metabolism by activating AMPK-mediated signaling, and, hence, could have a therapeutic role in the management and treatment of obesity.

## 1. Introduction

Anthocyanins are water-soluble phytochemicals present in pigmented vegetables, fruits, flowers, leaves, and seeds [1]. A total of 635 structurally different anthocyanins have been reported and classified into six common types [2,3]. Anthocyanins have potential anti-oxidative, antimicrobial, anti-inflammatory, anti-aging, anti-diabetic, anti-cancerous, and anti-obesity bioactivities [3,4,5,6]. Recent studies have reported that anthocyanin-rich fruits, including cranberries [7], polyphenolic extract [8], and *Vitis coingnetiae* (Meoru in Korea) [9] have potential anti-obesity activities, while a review article recently summarized the anti-obesity effects of berries. However, the data from these studies are inconsistent [10]. 

Two common anthocyanins, known as delphinidin-*3*-*O*-glucoside (D3G) and cyanidin-*3*-*O*-glucoside (C3G), are the glycosylated forms of Dp and Cy, respectively [2,11]. Cy is found in 50% of plants and its anti-obesity, anti-oxidant, and anti-inflammatory potential has been documented [12,13,14,15]. Cy has been shown to increase the differentiation of 3T3-L1 preadipocytes by decreasing the relative expression of a carbohydrate response element-binding protein (ChREBP) [16]. A study by Kim et al. revealed that the anthocyanins, D3G, C3G, and petunidin-3-*O*-glucoside (P3G) from black soybean reduced 3T3-L1 differentiation by decreasing the expression of peroxisome proliferator-activated receptor gamma (PPARγ) [17]. Previous studies have focused on the anti-obesity effects of C3G and anthocyanin-rich foods. However, a recent in vitro study evaluated the anti-adipogenic, anti-inflammatory, and anti-diabetic properties of anthocyanin-rich water extracts (PMWs) in 3T3-L1 preadipocytes and RAW 264.7 macrophages. PMWs prepared from 20 purple-colored maize genotypes were found to be particularly rich in C3G, peonidin-*3-O*-glucoside (P3G), and pelargonidin-*3-O*-glucoside (Pr3G), their acylated forms, and other phenolics [18]. The main contributors to the bioactivities of the PMWs were C3G, P3G, and their acylated forms. Furthermore, phenolic acids such as vanillic acid and protocatechuic acid showed anti-adipogenic activities, while quercetin, rutin, and luteolin displayed anti-inflammatory and anti-diabetic activities [18]. 

Dp is the second most common anthocyanidin (12 %), which is present in pigmented fruits and vegetables [10,19] and has antioxidant [20], anti-inflammatory [19], anti-atherosclerosis [21], anti-cancer [22], muscular atrophy prevention [23], and bone-protective [24] bioactivities. However, few studies have examined the beneficial effects of D3G on obesity. A recent study by Rahman et al. elucidated the inhibitory effects of Dp on 3T3-L1 preadipocytes differentiation by activating Wnt signaling [25], while D3G has been shown to suppress lipid accumulation in HepG2 cells [26]. 

During the 21st century, obesity has become an increasing global epidemic health concern since it is a risk factor for various complications, including heart diseases, type 2 diabetes mellitus (T2DM), hypertension, and cancer [2]. A strict diet, exercise regime, behavioral adaptation, and drug therapy are generally used to control body weight. However, these approaches are often unsuccessful in the long term since obese people struggle to manage them properly [27]. Numerous conventional anti-obesity drugs have been approved and are currently in use. However, detrimental side effects limit their utilization. Therefore, plant-based bio-actives have been studied as a potential strategy for counteracting weight gain, according to Bordoni et al. Bioactive compounds such as anthocyanins, β-glucan, catechins, and n-3 long chain PUFA (LCPUFA) show good anti-obesity activities [28]. 

Reports have shown that activation of the adenosine monophosphate-activated protein kinase (AMPK)-mediated pathway is associated with cellular energy homeostasis, and is suggested to have anti-adipogenic effects by reducing the expression of transcription factors such as PPARγ, CCAAT/enhancer-binding protein alpha (C/EBPα), and a sterol regulatory element-binding transcription factor 1 (SREBP1) [29]. AMPK also phosphorylates acetyl-CoA carboxylase (ACC), which is a fatty acid oxidation enzyme. This action reduces its levels [30]. Consequently, bioactive compounds that prevent preadipocyte differentiation and fat accumulation, and induce AMPK-mediated signaling, could be crucial for treating obesity. Since little information is available regarding the anti-obesity effects of D3G, this study aimed to describe these effects, which found that D3G inhibited adipocyte cell differentiation and lipid metabolism in 3T3-L1 adipocytes and primary white adipocytes (PWATs). Furthermore, we attempted to elucidate the mechanism underlying the anti-adipogenic activity of D3G. To our knowledge, this is the first study to describe the anti-obesity activity of D3G and its effects on AMPK-mediated signaling. 

## 2. Results

### 2.1. Effects of D3G on Viability and Intracellular Lipid Accumulation 

The effect of D3G on cell viability was measured using a Cell Counting Kit-8 (CCK-8) assay. No cytotoxicity was observed at a D3G concentration of 100 µM for 72 h (Figure 1). To assess the effect of D3G on lipid accumulation, cells were treated with D3G at concentrations of 25, 50, and 100 µM and stained with Oil Red O. As shown in Figure 2A,B, D3G significantly reduced the accumulation of intracellular lipids in a dose-dependent manner.

### 2.2. Effect of D3G on Adipogenesis

To investigate the role of adipocyte-specific transcription factors on 3T3-L1 preadipocyte differentiation, we studied the effects of D3G on PPARγ, SREBP1, and C/EBPα expression at both the transcriptional level and the translational level. As shown in Figure 3A–D, qRT-PCR analysis revealed a dose-dependent reduction in the mRNA and protein expression of PPARγ, SREBP1, and C/EBPα. The relative protein expression of PPARγ, SREBP1, and C/EBPα in the D3G-treated differentiating cells is presented in Figure 3E. These results show that D3G reduced the expression of important adipogenic transcription factors in 3T3-L1 adipocytes.

### 2.3. Effect of D3G on Fatty Acid Metabolism-Associated Genes

We examined the effects of D3G on lipogenesis and fat oxidation in 3T3-L1 adipocytes. Fatty acid synthase (FAS) expression was significantly downregulated by D3G treatment (Figure 4A,D), while 100 µM D3G significantly upregulated the expression of silent mating type information regulation 2 homolog 1 (SIRT1) and carnitine palmitoyltransferase -1 (CPT-1) (Figure 4B–D).

### 2.4. Effect of D3G on AMPK in 3T3-L1 Preadipocytes and PWATs

The AMPK pathway plays an essential role in regulating the early phase of adipogenesis. In this study, the expression of AMPK and acetyl-CoA carboxylase (ACC) was analyzed using Western blotting. As shown in Figure 5A,B, D3G significantly upregulated the expression of activated AMPK (p-AMPK/AMPK) and ACC (p-ACC/ACC). 

We also investigated the effect of D3G (25, 50, and 100 µM) on PWATs. D3G treatment significantly increased the p-AMPK/AMPK ratio (Figure 6A) and p-ACC/ACC expression (Figure 6B). These results indicate the activation of the AMPK-mediated signaling pathway.

To confirm whether AMPK activation was involved in the anti-adipogenic effects of D3G, PWATs were treated with an AMPK activator, 5-aminoimidazole-4-carboxamide ribonucleotide (AICAR, 10 µM), and the inhibitor dorsomorphin (10 µM). Western blot analysis revealed that D3G upregulated p-AMPK/AMPK and p-ACC/ACC expression in AICAR-treated and dorsomorphin-treated PWATs (Figure 7A,B), which suggests that 100 µM D3G induces AMPK activation. Furthermore, AMPK activation decreased PPARγ, SREBP1, C/EBPα, and FAS expression and increased CPT-1 and SIRT1 expression. In this study, D3G reduced adipogenesis and lipogenesis by activating AMPK, which regulated energy homeostasis and phosphorylated ACC in the rate-limiting step of lipogenesis.

## 3. Discussion

Obesity and obesity-related metabolic disturbances have increased at an alarming rate worldwide. Due to the adverse side effects of the currently-available anti-obesity drugs, there is a demand for safe, cost-effective, plant-derived, or food-derived, natural alternatives. Tsuda et al. were the first to report the protective effects of anthocyanins against lipogenesis in the body [31]. Since then, a number of studies have demonstrated the anti-obesity effects of plant-based bioactive compounds, including phenolics, flavonoids, and anthocyanins. However, their findings have been controversial [2,10,32,33]. It is, therefore, necessary to investigate the individual and synergistic effects of various anthocyanins. 

This study examined the effects of D3G on the differentiation of 3T3-L1 preadipocytes and PWATs. Adipocyte differentiation and lipid accumulation are crucial signs of obesity development [34]. Multiple studies have suggested that the differentiation of preadipocytes into adipocytes depends on a highly regulated multi-step cascade of transcription factors, among which PPARγ and C/EBPα are two key regulators [35,36]. PPARγ is a regulator of the lipid anabolic pathway. Its overexpression increased the size and number of lipid molecules in differentiating 3T3-L1 preadipocytes, whereas C/EBPα regulates many adipocyte genes. SREBP1 promotes preadipocyte differentiation and is associated with the increased expression of adipogenic genes [9,35,37]. In this study, real-time PCR and Western blotting analysis revealed that D3G downregulated the expression of PPARγ, C/EBPα, and SREBP1 in a concentration-dependent manner. Oil Red O staining showed reduced lipid growth in 3T3-L1 preadipocytes, which suggests that D3G may have anti-adipogenic effects. Furthermore, Dy treatment downregulated PPARγ and C/EBPα expression and inhibited 3T3-L1 preadipocyte differentiation and inhibitory effects, which may be related to the activation of the Wnt/*β*-catenin pathway [25]. Our results support the anti-differentiation effects of cranberries, which were shown to decrease the expression of PPARγ, SREBP1, and C/EBPα in 3T3-L1 preadipocytes [7].

FAS is the primary enzyme of lipogenesis and is expressed mainly in adipose tissue, the liver, and mammary glands [38]. Reduced FAS expression promotes the inhibition of preadipocyte differentiation and lipid accumulation [38]. We demonstrated that D3G downregulated FAS expression at the translational level. SREBP1 regulates FAS by promoting the transactivation of its enhancers [9]. Therefore, reduced FAS and SREBP1 expression is further evidence of the anti-adipogenic effects of D3G. According to Kowalska et al., the inhibitory activities of the anthocyanin-rich multi-berry (bilberry, chokeberry, cranberry, and raspberry) can be attributed to the downregulation of PPARγ, C/EBPα, SREBP1, and FAS gene expression [8].

SIRT1 and AMPK play important roles in the regulation of mitochondrial metabolism [39]. In this study, D3G significantly upregulated SIRT1 expression, which is a finding supported by previous research. This demonstrated that enhanced SIRT1 expression increases the lysis of lipid molecules and decreases fat accumulation, which negatively modulates adipogenesis in 3T3-L1 cells [40].

CPT-1 is an essential protein associated with increased mitochondrial activity and fatty acid oxidation [41]. In this study, D3G significantly increased the expression of CPT-1 in a concentration-dependent manner, which suggests that D3G may have a role in fatty acid oxidation. 

AMPK is a serine/threonine protein kinase and a crucial regulator of energy homeostasis in humans [9]. Previous evidence suggests that the level of p-AMPK (activated AMPK) is decreased during preadipocyte differentiation and adipogenesis [29]. AMPK activation can reduce obesity by increasing the expression of lipid metabolic genes at the transcriptional and translational level [29]. Our results showed that D3G significantly increased the expression of activated AMPK (p-AMPK), which increased phosphorylated ACC (p-ACC) expression in a dose-dependent manner and decreased the production of ACC, which is a key enzyme in fatty acid synthesis and degradation. This results in truncated fatty acid synthesis and elongation [30]. Another study revealed that active AMPK inhibits preadipocyte differentiation by reducing C/EBPα, PPARγ, and SREBP1 expression [42]. 

The effects of D3G on PWATs were also investigated. Similarly, D3G upregulated the expression of p-AMPK/AMPK and p-ACC/ACC. To further evaluate the D3G-mediated activation of AMPK, PWATs were treated with an AMPK activator (AICAR) and inhibitor (dorsomorphin). As shown in Figure 7, D3G enhanced the expression of p-AMPK/AMPK and p-ACC/ACC in the presence of both the activator and inhibitor, which demonstrates that D3G plays an important role in the activation of AMPK-mediated signaling. 

Until recently, the majority of anti-obesity effects had been attributed to C3G or a mixture of anthocyanins, and little was known about the anti-obesity activity of D3G. However, data from a human study alongside in vitro analysis showed that the plasma lipid-reducing effect of the anthocyanin-rich black soybean testa (BTT) was due to D3G rather than C3G. Furthermore, BTT exhibited a higher concentration of D3G than C3G [43]. Therefore, D3G appears to be an important anthocyanin and further studies are required to reveal its full health-promoting potential. This is the first study describing the anti-adipogenic effects of D3G in 3T3-L1 adipocytes and PWATs, and, taken together, our findings reveal that D3G could act as an important anti-adipogenic bioactive entity.

## 4. Materials and Methods

### 4.1. Materials 

D3G (chemical formula C_21_H_21_O_12_Cl, >97% purity) was purchased from Polyphenols AS (Sandnes, Norway). Dorsomorphin, AICAR, indomethacin, 3-isobutyl-1-methylxanthine (IBMX), and dexamethasone (DEX) were obtained from Sigma-Aldrich (St. Louis, MO, USA). Dulbecco’s modified Eagle’s medium (DMEM), bovine calf serum (BCS), trypsin-EDTA, and fetal bovine serum (FBS) were procured from Thermo Fisher (San Jose, CA, USA). CCK-8 was purchased from Dojindo Molecular Technologies (Rockville, MD, USA). Insulin was purchased from Thermo Fisher (San Jose, CA, USA). Random hexamers and the reverse transcriptase enzyme mix (GoScriptTM) were obtained from Promega (Madison, WI, USA). Monoclonal antibodies against C/EBPα, SREBP1, PPARγ, FAS, ACC, SIRT1, p-ACC, AMPK, p-AMPK, CPT-1, and horseradish peroxidase (HRP)-coupled anti-rabbit or anti-mouse secondary antibodies were purchased from Abcam (Cambridge, UK). Total RNA was extracted using an easy-spin^TM^ Total RNA Extraction Kit (iNtRON Biotechnology, Seongnam-si, Korea) and a PRO-MEASURE^TM^ protein measurement solution, protein lysis buffer, and a Western blot detection system were purchased from iNtRON Biotechnology (Seongnam-si, Korea). The polyvinylidene difluoride (PVDF) membrane was procured from Merck (Burlington, Massachusetts, USA). 

### 4.2. Cell Culture, Animal Preparation, and Differentiation

3T3-L1 preadipocytes were acquired from the American Type Culture Collection (ATCC, CL173, Manassas, VA, USA) and cultured in DMEM containing 10% BCS and 1% antibiotics in a 5% CO_2_ incubator at 37 °C. Differentiation was initiated by treating the cells with a differentiation medium (0.5 mΜ IBMX, 1 μΜ DEX, and 5 μg/ml insulin in DMEM with 10% FBS) for three days.

Stromal vascular fractions (SVFs) were obtained from the subcutaneous fat tissue of five-week to six-week-old male C57BL/6 mice, as described previously [44]. Isolated fat pads were crushed using scissors and digested with collagenase type II (Sigma, St. Louis, MO, USA) enzyme in a water bath at 37 °C. After 90 min, the mixture was filtered using 40-μm cell strainers (SPL Life Science, Pocheon-si, Gyeonggi-do, Korea). The filtered SVF cells were washed with phosphate buffered saline (PBS), centrifuged at 300× *g* for 7 min, and cultured in DMEM supplemented with 10% BCS and 1% antibiotics to obtain confluence. Differentiation was then initiated by incubating cells in a differentiation induction medium (100 μΜ indomethacin, 0.5 mΜ IBMX, 1 μΜ DEX, and 5 μg/ml insulin).

All experiments were approved by the guidelines for the care and use of laboratory animals of Gachon University (reference number: GIACUC-R2018016).

To evaluate the effects of D3G on adipocyte differentiation, 3T3-L1 preadipocytes, and PWATs were incubated with D3G (25, 50, and 100 μg/ml) for 7 days. Differentiated cells without D3G were used as a positive control (0) and undifferentiated cells were used as a negative control. 

### 4.3. Cell Viability Assay 

To evaluate the effect of D3G on viability, 3T3-L1 preadipocytes were seeded in a 96-well plate at a density of 1 × 10^4^ cells/well, treated with different concentrations (20, 50, 100, 200, and 500 µM) of D3G for 24, 48, and 72 h, and then, a CCK-8 kit was used according to the manufacturer’s instructions. The resulting absorbance was detected at 450 nm using a plate reader (Biotek Inc., Winooski, VT, USA). The results were calculated as a percentage of cell viability and 3 replicates were performed for each treatment. 

### 4.4. Quantification of Accumulated Lipids by Oil Red O Staining

After adipocyte differentiation, the control and D3G-treated cells were washed with PBS three times and fixed with 10% formalin for 5 min at room temperature (RT). The formalin was removed with 60% isopropanol, the wells were dried, and a working solution of Oil Red O (w/v, 60% isopropanol, 40% distilled water) was added to each well, incubated at RT for 1 h, and washed three times. Representative images of the stained cells were captured using an inverted optical microscope (Nikon Eclipse, Shinagawa, Tokyo, Japan). Lastly, the dye was eluted for quantitative analysis with 100% isopropanol and the absorbance was measured at 500 nm. Results were expressed as a percentage of the Oil Red O stained material compared to the control.

### 4.5. RNA Extraction and cDNA Synthesis 

Total RNA was isolated from the D3G-treated cells using an RNA isolation kit, according to the manufacturer’s instructions. RNA (50 ng) was converted into cDNA using a PCR machine (TaKaRa Bio, Kusatsu, Shiga, Japan) under the following conditions, which include initial incubation at 70 °C for 5 min followed by primer annealing at 25 °C for 5 min, extension at 42 °C for 60 min, and inactivation at 70 °C for 15 min.

### 4.6. Gene Expression Analysis Using a Quantitative Polymerase Chain Reaction (qPCR)

Quantification of the gene expression of 3T3-L1 adipocytes and PWATs was performed using QuantStudio3 RT-PCR system (Applied Biosystems, Thermo Fisher Scientific, San Jose, CA, USA). Amplification was performed for each target gene using 5.5 µl cDNA, 1 µl each primer (10 µM, forward and reverse), 10 µl 2X TB green mix (TaKaRa Bio, Kusatsu, Shiga, Japan), and 2.5 µl deionized water. The expression levels of the target genes were normalized to β-actin. Table 1 shows the primer sequences used for qPCR.

### 4.7. Protein Quantification and Immunoblot Analysis

Before harvesting, D3G-treated and control cells were washed once with PBS and extracted using a protein lysis buffer supplemented with inhibitor cocktails (protease and phosphatase). After scraping, the cells were centrifuged at 13,000 rpm and 4 °C for 10 min to isolate the proteins. The protein extracts (30 μg) were diluted in 5X sample buffer and incubated at 95 °C for 5 min. Total protein (30 µl) from the 3T3-L1 adipocytes and white adipocytes were loaded and separated using sodium dodecyl sulfate-polyacrylamide gel electrophoresis (SDS-PAGE, 10%). Then electro-transferred onto PVDF membranes. The gel was then blocked for 1 h with skim milk (5%) at RT under shaking conditions and was then washed with PBST (1X PBS with Tween 20) three times. The membranes were immunoblotted with primary antibodies against C/EBPα, PPARγ, SREBP1, FAS, SIRT1, ACC, p-ACC, p-AMPK, AMPK, and CPT-1 at 4 °C overnight. Subsequently, the membranes were incubated with secondary antibodies in 5% skim milk for 1 h at RT. Reactive band signals were visualized using a Western blot detection system and Image Quant LAS 500 (GE Healthcare Bio-Sciences AB, Björkgatan, Uppsala, Sweden). 

### 4.8. AMPK Activation or Inhibition 

To examine the role of AMPK signaling on the anti-adipogenic effect of D3G in PWATs, they were treated with AICAR and dorsomorphin, an AMPK activator, and inhibitor, respectively. AICAR (10 μΜ) or dorsomorphin (10 μΜ) was added to the differentiation induction and maturation media until cells were harvested. Each experiment was repeated three times.

### 4.9. Statistical Analysis

All data are expressed as the mean ± SD. Three separate experiments were carried out, and each experiment was performed in triplicate. Statistical analysis was performed using GraphPad Prism 5.03 (GraphPad Software Inc., La Jolla, CA, USA) with one-way ANOVA and Tukey’s post-hoc test. Differences with *p* values less than 0.05 were considered statistically significant.

## 5. Conclusions

This study demonstrated that D3G downregulated the expression of adipogenesis and lipogenesis markers (PPARγ, C/EBPα, SREBP1, and FAS), inhibited lipid accumulation, and upregulated fatty acid metabolism gene expression (SIRT1 and CPT-1) in 3T3-L1 adipocytes. Consistently, D3G phosphorylated AMPK and ACC and increased their expression in PWATs by activating AMPK-mediated signaling. These findings suggest that D3G could have promising therapeutic applications in obesity treatment and management.

## Figures and Tables

**Figure 1 molecules-24-01848-f001:**
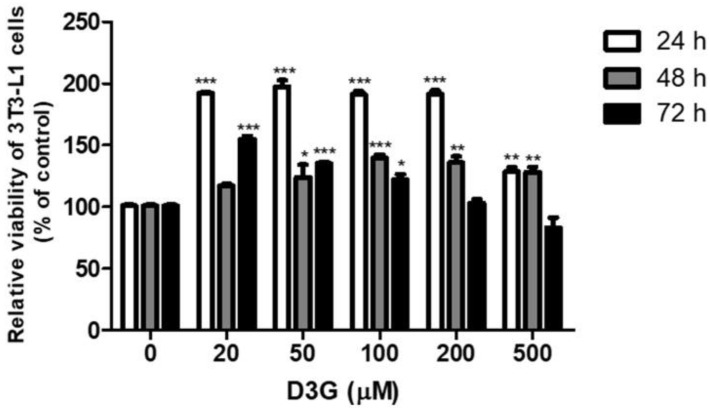
Effect of different delphinidin-3-*O*-*β*-glucoside (D3G) concentrations on the viability of 3T3-L1 preadipocytes. * *p* < 0.05, ** *p* < 0.01, and *** *p* < 0.001 vs. 0. All experiments were repeated at least three times and data represents the mean ± SD.

**Figure 2 molecules-24-01848-f002:**
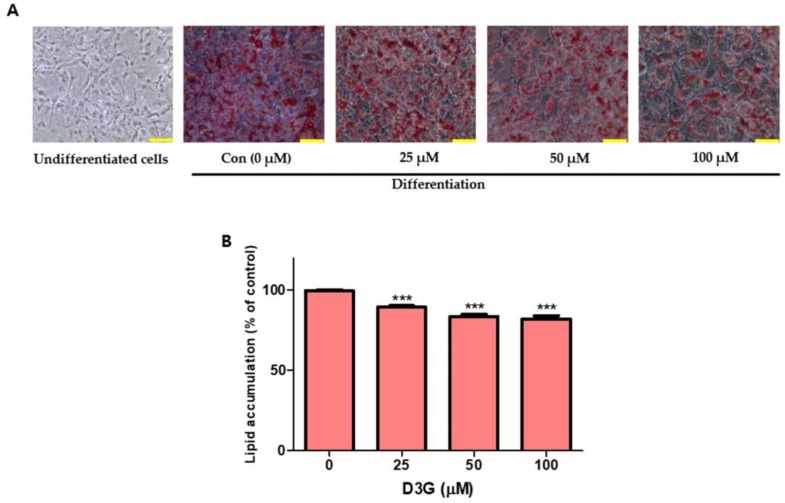
Effect of delphinidin-3-*O*-*β*-glucoside (D3G) on lipid accumulation in 3T3-L1 adipocytes. (**A**) Photomicrographs of lipid droplets in control (Con) and D3G (25, 50, and 100 µM)-treated cells measured by Oil Red O staining. (**B**) The percentage of lipid accumulation in D3G-treated and control cells. Data represent the mean ± SEM of three different experiments. *** *p* < 0.001 vs. 0. Scale bar (**A**) indicates 100 µm.

**Figure 3 molecules-24-01848-f003:**
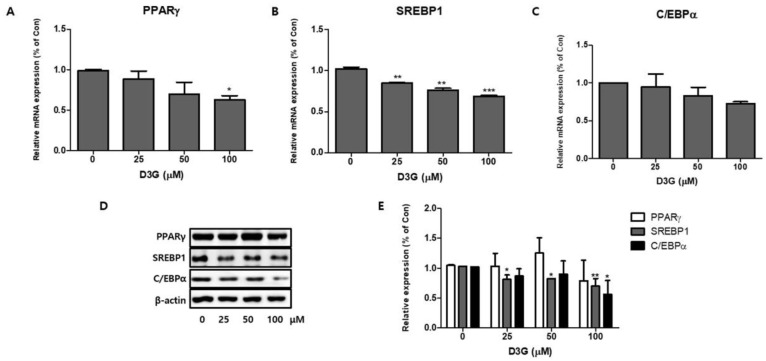
The effect of delphinidin-3-*O*-*β*-glucoside (D3G) treatment (25, 50, and 100 µM) on the mRNA expression of key adipogenic transcription factors: (**A**) peroxisome proliferator-activated receptor gamma (PPARγ), (**B**) sterol regulatory element-binding transcription factor 1 (SREBP1), and (**C**) CCAAT/enhancer-binding protein alpha (C/EBPα), as determined by quantitative real-time PCR. (**D**) Immunoblot analysis of PPARγ, SREBP1, and C/EBPα. (**E**) Relative protein expression of PPARγ, SREBP1, and C/EBPα in differentiating D3G-treated and control 3T3-L1 adipocytes. β-actin protein levels were used as an internal control. Data represent the mean ± SEM of three different experiments. * *p* < 0.05, ** *p* < 0.01, and *** *p* < 0.001 vs. 0.

**Figure 4 molecules-24-01848-f004:**
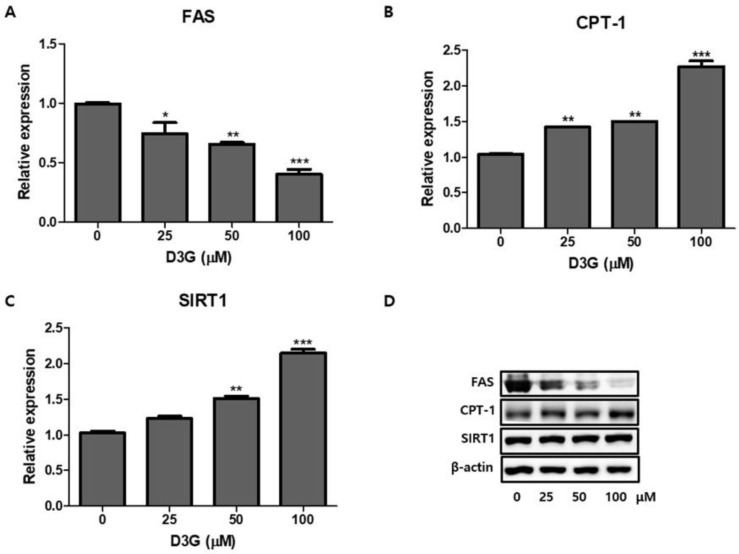
The effect of delphinidin-3-*O*-*β*-glucoside (D3G) treatment on the protein expression of (**A**) Fatty acid synthase (FAS), (**B**) carnitine palmitoyltransferase -1 (CPT-1), and (**C**) silent mating type information regulation 2 homolog 1 (SIRT1) in 3T3-L1 adipocytes. (**D**) Representative immunoblot image of FAS, CPT-1, and SIRT1 expression. β-actin protein levels were used as an internal control. * *p* < 0.05, ** *p* < 0.01, and *** *p* < 0.001 vs. 0. All experiments were repeated at least three times and data represent the mean ± SD.

**Figure 5 molecules-24-01848-f005:**
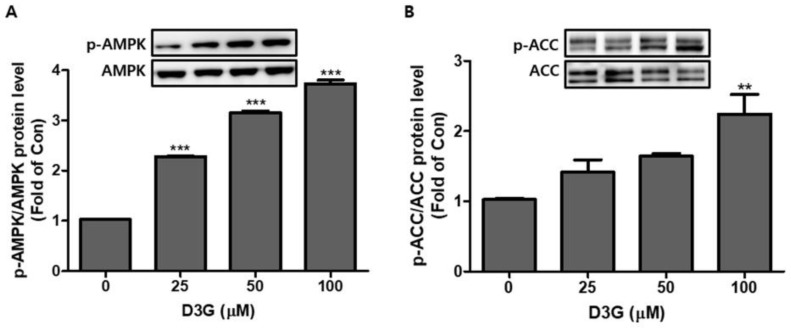
The effects of delphinidin-3-*O*-*β*-glucoside (D3G) treatment on the expression and phosphorylation of adenosine monophosphate-activated protein kinase (AMPK) and acetyl-CoA carboxylase (ACC) in 3T3-L1 adipocytes. Western blotting revealed the upregulated expression of (**A**) p-AMPK/AMPK and (**B**) p-ACC/ACC. ** *p* < 0.01, and *** *p* < 0.001 vs. 0. Each experiment was repeated three times and the data represents the mean ± SD.

**Figure 6 molecules-24-01848-f006:**
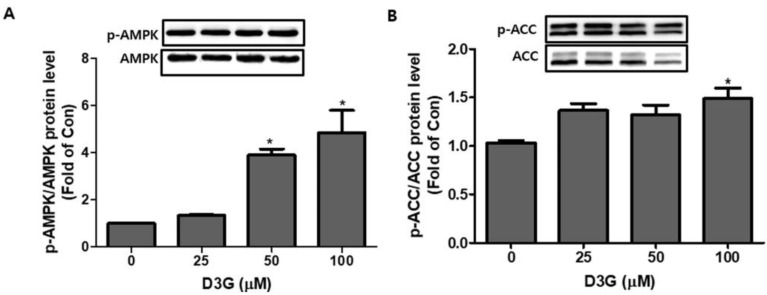
The effect of delphinidin-3-*O*-*β*-glucoside (D3G) treatment (25, 50, and 100 µM) on the expression and phosphorylation of adenosine monophosphate-activated protein kinase (AMPK) and acetyl-CoA carboxylase (ACC) in primary white adipocytes (PWATs). D3G treatment increased the expression of (**A**) p-AMPK/AMPK and (**B**) p-ACC/ACC in PWATs when compared to the control (0). * *p* < 0.05 vs. 0. Each experiment was repeated three times and the data represent the mean ± SD.

**Figure 7 molecules-24-01848-f007:**
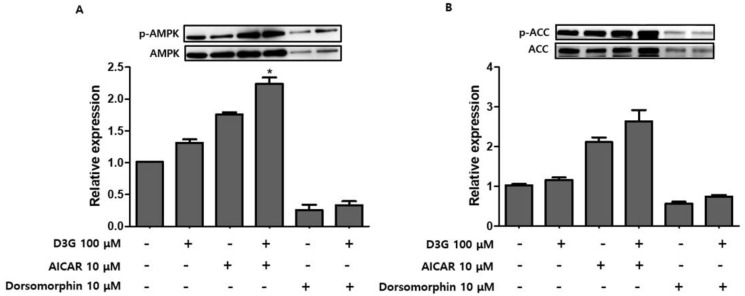
Effect of D3G on adenosine monophosphate-activated protein kinase (AMPK) activation in AMPK activator-treated and inhibitor-treated primary white adipocytes (PWATs). (**A**) p-AMPK/AMPK and (**B**) p-acetyl-CoA carboxylase (ACC)/ACC protein expression following D3G treatment in the presence of 5-aminoimidazole-4-carboxamide ribonucleotide (AICAR; 10 µM) and dorsomorphin (10 µM) in PWATs. Protein expression levels were measured by Western blotting. Each experiment was repeated three times and data represented the mean ± SD.

**Table 1 molecules-24-01848-t001:** Primer sequences for quantitative real-time PCR.

Gene	Forward (5′- 3′)	Reverse (5′- 3′)
C/EBPα	TTACAACAGGCCAGGTTTCC	GGCTGGCGACATACAGTACA
PPARγ	TTTTCAAGGGTGCCAGTTTC	AATCCTTGGCCCTCTGAGAT
SREBP1	TGTTGGCATCCTGCTATCTG	AGGGAAAGCTTTGGGGTCTA
β-actin	CTGTCCCTGTATGCCTCTG	ATGTCACGCACGATTTCC

C/EBPα: CCAAT/enhancer-binding protein alpha. PPARγ: peroxisome proliferator-activated receptor gamma. SREBP1: sterol regulatory element-binding transcription factor 1.

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
