# Peer review of "Anti-Adipogenic Effects of Delphinidin-3-O-β-Glucoside in 3T3-L1 Preadipocytes and Primary White Adipocytes"

_molecules, 2019, doi:10.3390/molecules24101848_

Round 1

Reviewer 1 Report

Do  

            The current study indicate that D3G attenuates adipogenesis and promotes lipid metabolism through the activation of the AMPK mediated pathway and hence may have a potential therapeutic role for obesity management and treatment. 

                Obesity and its associated diseases are a worldwide epidemic disease. Usual weight loss cures - as diets, physical activity, behavior therapy and pharmacotherapy - have been continuously implemented but still have relatively poor long-term success and mainly scarce adherence [1]. Anthocyanins (ACN), catechins, β-glucan (BG) and n-3 long chain PUFA (LCPUFA) are among the most promising candidates and have been considered as a strategy for the development of functional foods counteracting body weight gain [2]. Recently, anthocyanin-rich water extracts (PMWs) from 20 purple maize genotypes were evaluated in RAW 264.7 macrophages and 3T3-L1 adipocytes under different conditions [3]. Cyanidin-3-O-glucoside (C3G), pelargonidin-3-O-glucoside (Pr3G), peonidin-3-O-glucoside (P3G) and corresponding acylated forms were major anthocyanins in PMW, accompanied by ten tentatively identified non-anthocyanin phenolics [3]. Correlation studies showed that C3G, P3G, and derivatives, but not Pr3G and its acylated form contributed to the biological properties of PMW [3]. Besides anthocyanins, quercetin, luteolin, and rutin were the dominant anti-inflammatory and anti-diabetic components, in terms of down-regulating pro-inflammatory mediator production in inflamed macrophages and adipocytes, modulating diabetes-related key enzymes and improving insulin sensitivity in insulin-resistant adipocytes [3]. Quercetin and phenolic acids, especially vanillic acid and protocatechuic acid, were closely associated with anti-adipogenic properties of PMW via inhibition of the preadipocyte-adipocyte transition [3]. Another, it was suggested that GLP-1R may be a potential therapeutic target for the treatment of ALI [3].

                  Authors are kindly requested to emphasize the current concepts about these issues in the context of recent knowledge and the available literature. This article should be quoted in the References list.

References

Could the improvement of obesity-related      co-morbidities depend on modified gut hormones secretion? World J Gastroenterol.      2014 Nov 28; 20 (44): 16649-64. doi: 10.3748/wjg.v20.i44.16649.

The role of bioactives in      energy metabolism and metabolic syndrome. Proc Nutr Soc. 2019 Apr 10:1-11.      doi: 10.1017/S0029665119000545..

Relationship of phenolic      composition of selected purple maize (Zea mays L.) genotypes with their      anti-inflammatory, anti-adipogenic and anti-diabetic potential. Food Chem.      2019 Aug 15;289:739-750. doi: 10.1016/j.foodchem.2019.03.116.

Author Response

Dear reviewer,

Thank you for your constructive comments concerning our manuscript. We have revised the manuscript in accordance with your comments. We answered your questions or comments in detail in the following text.

Major comments:

The current study indicate that D3G attenuates adipogenesis and promotes lipid metabolism through the activation of the AMPK mediated pathway and hence may have a potential therapeutic role for obesity management and treatment.

                Obesity and its associated diseases are a worldwide epidemic disease. Usual weight loss cures - as diets, physical activity, behavior therapy and pharmacotherapy - have been continuously implemented but still have relatively poor long-term success and mainly scarce adherence [1]. Anthocyanins (ACN), catechins, β-glucan (BG) and n-3 long chain PUFA (LCPUFA) are among the most promising candidates and have been considered as a strategy for the development of functional foods counteracting body weight gain [2]. Recently, anthocyanin-rich water extracts (PMWs) from 20 purple maize genotypes were evaluated in RAW 264.7 macrophages and 3T3-L1 adipocytes under different conditions [3]. Cyanidin-3-O-glucoside (C3G), pelargonidin-3-O-glucoside (Pr3G), peonidin-3-O-glucoside (P3G) and corresponding acylated forms were major anthocyanins in PMW, accompanied by ten tentatively identified non-anthocyanin phenolics [3]. Correlation studies showed that C3G, P3G, and derivatives, but not Pr3G and its acylated form contributed to the biological properties of PMW [3]. Besides anthocyanins, quercetin, luteolin, and rutin were the dominant anti-inflammatory and anti-diabetic components, in terms of down-regulating pro-inflammatory mediator production in inflamed macrophages and adipocytes, modulating diabetes-related key enzymes and improving insulin sensitivity in insulin-resistant adipocytes [3]. Quercetin and phenolic acids, especially vanillic acid and protocatechuic acid, were closely associated with anti-adipogenic properties of PMW via inhibition of the preadipocyte-adipocyte transition [3]. Another, it was suggested that GLP-1R may be a potential therapeutic target for the treatment of ALI [3].

Could the improvement of obesity-related co-morbidities depend on modified gut hormones secretion? World J Gastroenterology. 2014 Nov 28; 20 (44): 16649-64. doi10.3748/wjg .v20.i44.16649.

The role of bioactives in energy metabolism and metabolic syndrome. Proc Nutr Soc. 2019 Apr 10:1-11.      doi: 10.1017/S0029665119000545..

Relationship of phenolic composition of selected purple maize (Zea mays L.) genotypes with their      anti-inflammatory, anti-adipogenic and anti-diabetic potential. Food Chem. 2019 Aug 15; 289:739-750. doi: 10.1016/j.foodchem.2019.03.116.

Thank you very much for your suggestion. We have revised the introduction section of the manuscript with suggested literature along with references as described below. We have highlighted the modification in red.

Line 45-53: Previous studies have focused on the anti-obesity effects of C3G and anthocyanin-rich foods; however, a recent in vitro study evaluated the anti-adipogenic, anti-inflammatory, and anti-diabetic properties of anthocyanin-rich water extracts (PMWs) in 3T3-L1 preadipocytes and RAW 264.7 macrophages. PMWs prepared from 20 purple-colored maize genotypes were found to be particularly rich in C3G, peonidin-3-O-glucoside (P3G), and pelargonidin-3-O-glucoside (Pr3G), their acylated forms, and other phenolics [18]. The main contributors to the bioactivities of the PMWs were C3G, P3G, and their acylated forms. Furthermore, phenolic acids such as vanillic acid and protocatechuic acid showed anti-adipogenic activities, while quercetin, rutin, and luteolin displayed anti-inflammatory and anti-diabetic activities [18].

[18] Zhang, Q.; de Mejia, E. G.; et al. Relationship of phenolic composition of selected purple maize (Zea mays L.) genotypes with their anti-inflammatory, anti-adipogenic and anti-diabetic potential. Food Chem. 2019, 289:739-750. doi: 10.1016/j.foodchem.2019.03.116.

Line 62-64: A strict diet, exercise regime, behavioral adaptation, and drug therapy are generally used to control body weight; however, these approaches are often unsuccessful in the long term since obese people struggle to manage them properly [27].

[27] Finelli, C.; Padula, M.C.; et al. Could the improvement of obesity-related co-morbidities depend on modified gut hormones secretion? World J Gastroenterology. 2014, 20 (44): 16649-64. doi10.3748/wjg .v20.i44.16649.

Line 66-68: Numerous conventional anti-obesity drugs have been approved and are currently in use; however, detrimental side effects limit their utilization. Therefore, plant-based bioactives have been studied as a potential strategy for counteracting weight gain; according to Bordoni et al., bioactive compounds such as anthocyanins, β-glucan, catechins, and n-3 long chain PUFA (LCPUFA) show good anti-obesity activities [28].

[28] Bordoni, A.; Boesch, C.; et al. The role of bioactives in energy metabolism and metabolic syndrome. Proc Nutr Soc. 2019, 1-11. doi: 10.1017/S0029665119000545..

Reviewer 2 Report

The authors investigated the anti-adipogenic effect of delphinidin-3-O-β-glucoside (D3G) in adipocytes. D3G lowered the accumulation of intracellular lipids, and the expression levels of the adipogenic and lipogenic genes. In contrast, the expression of SIRT1 and CPT-1 was increased. In addition, D3G promoted lipid metabolism through activation of AMPK and ACC. The manuscript is well organized, and could be read easily. However, there are concerns that should be addressed.

1. D3G has an ability to elevate the cell proliferation at 24 h in 3T3-L1 cells in Figure 1? Please comment to these results.

2. μM/ml should be μM.

3. Scale bars in Figure 2A are unclear. Please improve them.

4. In Figure 2A, “negative control” means “undifferentiated cells”? If so, “undifferentiated cells” are better than “negative control”.

5. In all figures, “Con” should be “0”.

6. What amount of proteins were loaded in each lane in Immunoblot analysis? Please indicate in Methods part or figure legend.

7. In Figures 5 and 6, how many times were replicated the studies? Please indicate.

8. What means “PWAT (primary white adipocytes)”? How were they prepared? It should be clearly written.

9. English should be improved. Some grammatical errors were found.

Author Response

Dear reviewer,

Thank you for your constructive comments concerning our manuscript. We have revised the manuscript in accordance with your comments. We answered your questions or comments in detail in the following text.

The authors investigated the anti-adipogenic effect of delphinidin-3-O-β-glucoside (D3G) in adipocytes. D3G lowered the accumulation of intracellular lipids, and the expression levels of the adipogenic and lipogenic genes. In contrast, the expression of SIRT1 and CPT-1 was increased. In addition, D3G promoted lipid metabolism through activation of AMPK and ACC. The manuscript is well organized, and could be read easily. However, there are concerns that should be addressed.

Major comments:

1. D3G has an ability to elevate the cell proliferation at 24 h in 3T3-L1 cells in Figure 1? Please comment to these results.

à Thank you for your question. Yes, we do agree that D3G has an ability to elevate the cell proliferation at 24 h in 3T3-L1 cells as shown in Figure 1. However, the differentiation process takes about 7 days. Therefore, we have checked the proliferation for 72 h and as mentioned no cytotoxicity was observed for 72 h. Each experiment was repeated three times.

2. μM/ml should be μM.

According to your advice, we have changed μM/ml to μM in the text and figure legends.

à Line 88: we have changed μM/ml to μM

à Line 95: we have changed μM/ml to μM in Figure 2 legend

à Line 108: we have changed μM/ml to μM in Figure 3 legend

à Line 142: we have changed μM/ml to μM in Figure 6 legend

3. Scale bars in Figure 2A are unclear. Please improve them.

à Thank you for your advice. According to your advice, we have improved the scale bar in Figure 2A.

4. In Figure 2A, “negative control” means “undifferentiated cells”? If so, “undifferentiated cells” are better than “negative control”.

à Thank you for your comments. We have changed the “negative control” with “undifferentiated cells” in Figure 2A.

5. In all figures, “Con” should be “0”.

à According to your advice, we have changed “con” with “0” in all figures.

6. What amount of proteins were loaded in each lane in Immunoblot analysis? Please indicate in Methods part or figure legend.

à Line 295-298: Thank you for your comments. As suggested, we have corrected the method section of the revised manuscript.

Total protein (30 µl) from the 3T3-L1 adipocytes and white adipocytes were loaded and separated using sodium dodecyl sulfate-polyacrylamide gel electrophoresis (SDS-PAGE, 10 %), then electrotransferred onto PVDF membranes.”

7. In Figures 5 and 6, how many times were replicated the studies? Please indicate.

à Thank you for your comments. Each experiment was repeated three times. We have    
  mentioned in the ‘Materials and Methods’ section and figure legends.

Line 309: Each experiment was repeated three times

Line 123-124: Figure 4 legend- All experiments were repeated at least three times and data represent the mean ± SD

Line135-136: Figure 5 legend- Each experiment was repeated three times and data represent the mean ± SD

Line144-145: Figure 6 legend- Each experiment was repeated three times and data represent the mean ± SD

Line158-159: Figure 7 legend- Each experiment was repeated three times and data represent the mean ± SD

8. What means “PWAT (primary white adipocytes)”? How were they prepared? It should be clearly written.

à We have included the relevant information and the reference in lines (242-249) in the Materials and Methods section. In studying adipogenesis, various markers have been used to characterize adipose precursors from stromal vascular fractions (SVFs) of WAT, containing multiple types of cells, including adipose precursors, endothelial cells, immune cells, and fibroblasts (Gulyaeva et al., 2018). Differentiated SVFs become WAT. PWAT (primary white adipocytes) is one of SVFs. Please kindly check the following sentences from the manuscript by Chruch et al.

It has been known that adipocyte precursors reside within WAT depots as culturing of the heterogeneous mixtures of WAT resident stromal cells, termed the stromal vascular fractions (SVF), results in the generation of lipid filled adipocytes”.

Gulyaeva O, Nguyen H, Sambeat A, Heydari K, Sul HS. Sox9-Meis1 Inactivation Is Required for Adipogenesis, Advancing Pref-1+ to PDGFRα+ Cells. Cell Rep. 2018 Oct 23;25(4):1002-1017.e4. doi: 10.1016/j.celrep.2018.09.086.

Church, C., Berry, R., and Rodeheffer M.S. Isolation and Study of Adipocyte Precursors. Methods Enzymol. 2014; 537: 31–46. doi: 10.1016/B978-0-12-411619-1.00003-3

9. English should be improved. Some grammatical errors were found.

à Thank you for your thoughtful review. Based on your feedback, we have performed the English proofreading from the language editing service.

Round 2

Reviewer 2 Report

The manuscript was improved. I have no further comment.